# Molecular detection of extended spectrum β-lactamase genes in *Escherichia coli* clinical isolates from diarrhoeic children in Kano, Nigeria

**Habeeb Kayode Saka**[1,2]*, **Silvia García-Soto**[3], **Nasir Tukur Dabo**[4], **Vicente Lopez-Chavarrias**[3], **Bashir Muhammad**[2], **Maria Ugarte-Ruiz**[3], **Julio Alvarez**[3,5]

**1** Nigerian Stored Products Research Institute, Kano, Nigeria, **2** Department of Microbiology, Bayero University Kano, Kano, Nigeria, **3** VISAVET Health Surveillance Centre, Universidad Complutense, Madrid, Spain, **4** Department of Biological Sciences, Bayero University Kano, Kano, Nigeria, **5** Departamento de Sanidad Animal, Facultad de Veterinaria, Universidad Complutense, Madrid, Spain

* saka.habeeb@yahoo.com

**Data Availability Statement:** All relevant data are within the manuscript and its Supporting Information files.

## Abstract

The increase in antimicrobial resistance in developed and developing countries is a global public health challenge. In this context β-lactamase production is a major contributing factor to resistance globally. The aim of this study was to determine the prevalence of phenotypic and genotypic extended spectrum β-lactamases (ESBLs) in 296 *E. coli* isolates recovered from diarrhoeic children younger than five years in Kano whose susceptibility profile against 7 antimicrobials had been determined. The *E. coli* isolates were subjected to double disc synergy test for phenotypic ESBLs detection and ESBL associated genes ($bla_{CTX-M}$, $bla_{TEM}$ and $bla_{SHV}$) were detected using conventional PCR. Phenotypically, 12.8% (38/296) *E. coli* isolates presented a ESBLs phenotype, with a significantly higher proportion in isolates from females compared with males (*P-value* = 0.024). $bla_{CTX-M}$ 73.3% and $bla_{TEM}$ 73.3% were the predominant resistance genes in the ESBLs positive *E. coli* (each detected in 22/30 isolates, of which 14 harboured both). In addition, 1/30 harboured $bla_{CTX-M}$ + $bla_{TEM}$ + $bla_{SHV}$ genes simultaneously. This study demonstrates the presence of ESBLs *E. coli* isolates in clinically affected children in Kano, and demonstrates the circulation of $bla_{CTX-M}$ and $bla_{TEM}$ associated with those phenotypes. Enactment of laws on prudent antibiotic use is urgently needed in Kano.

## Introduction

Hospital-based surveillance systems have reported an increase in the distribution of antibiotic resistant microorganisms in both developed and developing countries worldwide [1]. The ability to control infectious diseases, which includes diarrhoea caused by *Escherichia coli*, is currently endangered by this phenomenon, which has been declared as a global threat to public health [2]. Infections caused by resistant microorganisms often fail to respond to conventional antibiotics resulting in prolonged morbidity and higher mortality [2]. Even though antibiotics

**Funding:** The authors received no specific funding for this work.

**Competing interests:** The authors have declared that no competing interest exist.

are not recommended for treatment of diarrhoea caused by *E. coli*, diarrhoeagenic *E. coli* (DEC) [3, 4] can carry antimicrobial resistance genes that may be acquired through horizontal transfer from other resistant isolates within the same or other genus [5]. β-lactamase production in Gram negative bacteria is the most important contributing factor to β-lactam resistance [6], and has undergone evolution over time since its first appearance [7]. Newer β-lactamase-producing enterobacteriaceae have been isolated from clinical settings in different parts of the world [5] carrying factors such as plasmid-mediated cephamycinases, extended spectrum β-lactamases (ESBLs), and carbapenemases [8], and thus demonstrating the variability in the potential mechanisms behind this phenotype. ESBLs have the ability to hydrolyse penicillins, cephalosporins and monobactams, but not cephamycins and carbapenems [9]. They are inhibited by classical and newly developed β- lactamase inhibitors such as clavulanic acid, sulbactam, tazobactam, avibactam, relebactam or nacubactam, among others [5, 7, 10].

ESBLs producing *E. coli* are a frequent cause of community and hospital acquired infections, thus creating an increasing public health challenge [5], and are one of the leading causes of infections worldwide [11]. Acquired ESBLs emerged in the 1980s as derivatives of $bla_{TEM}$ (named after the patient Temoneira) and $bla_{SHV}$ (sulfhydryl reagent variable) enzyme types [5, 12]. The genes encoding these enzymes are plasmid mediated and therefore easily transmissible between bacteria of the same or different species. This phenomenon may be favoured by the extensive use of β-lactam antibiotics in human medicine [7]. The ESBLs genotype has been associated with hospital and community outbreaks in a pandemic manner [12].

Diarrhoeal disease is a major cause of morbidity and the second most important cause of mortality in children less than 5 years of age [13]. Diagnosis of the causative agents is very important for proper management and surveillance purposes. There is however limited epidemiological information on the presence and distribution of ESBLs producing *E. coli* from diarrheic children in Nigeria. Detection of ESBLs phenotypes and genotypes in a paediatric population within a geographic area is very important as emergence of ESBLs producing *E. coli* leads to carbapenems reliance as the alternative treatment option. Since there are no previous reports on molecular detection of ESBL associated genes among diarrhoeic children in Kano, here we aimed at investigating the prevalence of ESBLs phenotypes and the associated resistance genes in *E. coli* isolates from diarrhoeic children in this state in Nigeria.

## Materials and methods

### Study population

The study was conducted in Kano state, Nigeria. Specimens were collected from outpatient diarrhoeic children under five years attending three major hospital: Bichi General Hospital (BGH), Murtala Muhammad Specialist Hospital (MMSH) and Wudil General Hospital (WGH). Additional information of the sampled population is provided elsewhere [14]. Diarrhoea is defined as the passage of three or more loose or watery stool in 24 hours.

### Ethical consideration

Rectal swab specimens were obtained after obtaining informed consent from the parent and legal guardian of the children. Earlier on, ethical approval was granted by the Kano State Ministry of Health with Ref: MOH/Off/797/T.I/186.

### Sample collection, bacterial isolation and antimicrobial susceptibility

Two hundred and ninety six *E. coli* isolates recovered from swabs from children with diarrhoea were obtained as described elsewhere [14]. Results from the antimicrobial susceptibility tests

against Cefuroxime (CXM) 30 μg, Ceftazidime (CAZ) 30 μg, Cefotaxime (CTX) 30 μg, Amoxi-cillin-clavulanic acid (AMC) 30 μg, Gentamicin (CN) 10 μg, trimethoprim-sulfamethoxazole (SXT) 25 μg, and ciprofloxacin (CIP) 5 μg in all isolates has been previously reported [14]. Phe-notypic ESBLs screening and confirmation was carried out on all 296 *E. coli* as recommended by CLSI [15]. The screening was carried out using Ceftazidime 30 μg disc (*Oxoid*, *UK*): a zone of inhibition ≤ 22mm was considered suggestive of ESBLs production and positive isolates were further investigated using the double disc synergy test with a combination of three antibi-otic discs (ceftriaxone, amoxicillin-clavulanic acid and ceftazidime). A ≥ 5 mm increase in the inhibition zone for either antibiotic towards the amoxicillin-clavulanic acid with a dumbbell shape was considered indicative of an ESBLs phenotype. In addition, Tetracycline (TET) 30 μg and Imipenem 10 μg antibiotics (*Oxoid UK*) were tested on the ESBLs positive *E. coli*. Results were recorded as susceptible, intermediate and resistant according to the reference zone of inhibition of each antibiotic according to CLSI [15]. Isolates expressing ESBLs phenotype were preserved in Tryptic soy broth (TSB) (*Biomerieux France*) supplemented with 20% glycerol at -80˚C until further testing.

## DNA extraction

All *E. coli* isolates that were previously preserved in TSB+20% glycerol were cultured onto Eosin Methylene Blue (EMB) agar at 37˚C for 18–24 hours. One loop full of *E. coli* from the EMB plates was suspended in about 2 ml of sterile distilled water in an eppendorf tube; the bacterial suspension was boiled at 100˚C in a water bath for 10 minutes and centrifuged at 13,000 rpm for 1minute as previously described [16]. The supernatant was used as DNA template for PCR.

## Detection of ESBLs associated genes by PCR

The molecular detection of ESBLs associated genes ($bla_{SHV}$, $bla_{TEM}$ and $bla_{CTX-M}$) was carried out using conventional multiplex PCRs in the majority of the ESBL presumptive *E. coli* isolates. Primers used were previously described by Monstein *et al.* [17]. The primer mix for the detec-tion of ESBLs associated genes was prepared in an eppendorf tube by adding 210 μl of ultrapure water, 5μl uidA- Forward primer + 5μl uidA–Reverse primer, 5μl SHV—Forward primer + 5μl SHV–Reverse primer, 5μl CTX-M–Forward primer + 5μl CTX-M–Reverse primer, 5μl TEM–Forward primer + 5μl TEM–Reverse primer. Eppendorf tubes containing 22 μl of the reaction mixture were used for the PCRs (8 μl of water (Biorad) + 10 μl Mastermix (*Bio-Rad*) + 2μl of the mix primer + 2 μl of DNA). The ultrapure PCR cycling conditions were as follows: initial dena-turation for 15 seconds at 95˚C, 30 cycles of denaturation at 95˚C for 30 seconds, annealing at 60˚C for 1 minute 30 seconds, elongation at 72˚C for 2 minutes and final elongation at 72˚C for 10 minutes. Biorad MyCycler PCR thermal cycler was used to run the PCR cycles. The post amplification products were analysed using 2% agarose gel electrophoresis. Gel Doc XR+ Imag-ing system (*Bio-Rad*) was used in viewing the gel after exposure to UV light.

Isolates positive in the PCR for detection of $bla_{TEM}$ genes were subjected to another PCR for amplification and sequencing of the complete gene in order to identify the subtypes pres-ent. In this case the primer mix consisted of 5μl TEMSEQ forward primer + 5μl TEMSEQ reverse primer [18] + 240μl ultrapure water, and the remaining steps as described above. PCR cycling conditions consisted of initial denaturation for 15 seconds at 95˚C, 40 cycles of dena-turation at 95˚C for 30 seconds, annealing at 50˚C for 1 minute 30 seconds, elongation at 72˚C for 1 minute and final elongation at 72˚C for 10 minutes. Successfully amplified PCR products were purified using Illustra ExoProStar 1-Step, and sequenced. Sequences were analyzed using Bioedit [19] and MEGA X [20] and identified by comparing them with the NCBI database using BLAST [21].

## Results

An ESBL phenotype was detected in 38 (12.84%) of the collection of 296 *E. coli* retrieved from the rectal swab specimens. ESBLs phenotype was significantly (chi-square test, *P*-value = 0.024) more common among *E. coli* recovered from female patients (17.83%, 23/129) compared with samples from males (8.98%, 15/167). Thirty out of the 38 phenotypic ESBLs positive *E. coli* (due to inability to recover eight isolates) were screened for ESBLs associated genes ($bla_{SHV}$, $bla_{TEM}$ and $bla_{CTX-M}$): all 30 isolates tested positive for at least one of the three resistance associated gene targets; $bla_{CTX-M}$ and $bla_{TEM}$ were detected in 73.3% (22/30), and $bla_{SHV}$ was detected in 6.66% (2/30) of the *E. coli* screened. The $bla_{TEM}$ sequences of 16/22 isolates carrying this gene were obtained, revealing two variants: 13 sequences matched perfectly with a previously published $bla_{TEM-1}$ sequence (genbank NG050145.1), and the other three had just one single nucleotide polymorphism in position 396).

The susceptibility patterns of the ESBLs producing *E. coli* and the resistance pattern of *E. coli* based on the presence of the ESBLs associated genes are presented in Tables 1 and 2. All the ESBLs positive *E. coli* were resistant to tetracycline (except for only 2 isolates in the phenotypic group with intermediate resistance) and susceptible to imipenem, while intermediate levels of resistance were found for the rest of the antimicrobials. Proportion of resistance to all antimicrobials in isolates harbouring ESBL genes was very similar or identical to that found in the phenotypic-positive ESBLs isolates, although the percentage of isolates harbouring a MDR resistance profile (simultaneous resistance to three or more antimicrobial families) was higher among genotypically confirmed isolates (90.0%) compared with total ESBL presumptive isolates (78.9%) (Table 1). When comparing isolates with either $bla_{CTX-M}$ or $bla_{TEM}$ (n = 7 in both cases), a higher proportion of resistant isolates for all antimicrobials except AMC and SXT were found for those harbouring the former (Table 1). Interestingly, the simultaneous presence of both $bla_{CTX-M}$ and $bla_{TEM}$ led to higher proportion of resistance compared with the presence of either of the genes in several cases (Table 2).

## Discussion

The spread of plasmid-encoded extended-spectrum β-lactamase (ESBLs) genes [5], conferring resistance to third-generation cephalosporins including aztreonams [9] is considered a major contributor to the ongoing emergence of antimicrobial resistance. The proportion of *E. coli*

**Table 1. Antibiotic susceptibility pattern of ESBLs producing *E. coli*.**

| Antibiotics | ESBLs Phenotypic (N = 38) | | | | ESBLs Genotypic (N = 30) | | | |
|---|---|---|---|---|---|---|---|---|
| | S | I | R | MDR | S | I | R | MDR |
| | n (%) | n (%) | n (%) | n (%) | n (%) | n (%) | n (%) | n (%) |
| Cefuroxime | 2 (5.3) | 9 (23.7) | 27 (71.1) | | 1 (3.3) | 7 (23.3) | 22 (73.3) | |
| Cefotaxime | 3 (7.9) | 7 (18.4) | 28 (73.7) | | 3 (10.0) | 5 (16.7) | 22 (73.3) | |
| Amox-Clav | 3 (7.9) | 7 (18.4) | 28 (73.7) | | 1 (3.3) | 5 (16.7) | 24 (80.0) | |
| Ceftazidime | 4 (10.5) | 8 (21.1) | 26 (68.4) | | 3 (10.0) | 5 (16.7) | 22 (73.3) | |
| Ciprofloxacin | 20 (52.6) | 5 (13.2) | 13 (34.2) | 30 (78.9) | 14 (46.7) | 5 (16.7) | 11 (36.7) | 27 (90.0) |
| Gentamycin | 29 (76.3) | 0 (0.0) | 9 (23.7) | | 22 (73.3) | 0 (0.0) | 8 (26.7) | |
| Cotrimoxazole | 3 (7.9) | 0 (0.0) | 35 (92.1) | | 3 (10.0) | 0 (0.0) | 27 (90.0) | |
| Tetracycline | 0 (0.0) | 2 (6.7) | 36 (94.7) | | 0 (0.0) | 0 (0.0) | 30 (100.0) | |
| Imipenem | 38 (100) | 0 (0.0) | 0 (0.0) | | 30 (100) | 0 (0.0) | 0 (0.0) | |

S-Susceptible, I-Intermediate, R-Resistant, MDR- Multi-Drug Resistant, Amox-Clav- Amoxicillin-clavulanic acid.

**Table 2. Antibiotics resistance pattern according to detected associated ESBLs genes.**

| | Antibiotics | CXM | CTX | CAZ | AMC | CIP | CN | SXT | MDR |
|---|---|---|---|---|---|---|---|---|---|
| Associated Gene | N(%) | n(%) | n(%) | n(%) | n(%) | n(%) | n(%) | n(%) | n(%) |
| $bla_{CTX-M}$ Only | 7(23.33) | 5(71.4) | 5(71.4) | 5(71.4) | 5(71.4) | 4(57.1) | 2(28.6) | 6(85.7) | 5(71.4) |
| $bla_{TEM}$ Only | 7(23.33) | 2(28.6) | 3(42.9) | 3(42.9) | 6(85.7) | 0(0.0) | 2(28.6) | 6(85.7) | 6(85.7) |
| $bla_{SHV}$ Only | 1(3.33) | 1(100.0) | 0(0.0) | 0(0.0) | 0(0.0) | 0(0.0) | 0(0.0) | 1(100.0) | 1(100.0) |
| $bla_{CTX-M}+bla_{TEM}$ | **14(46.67)** | **13(92.9)** | **13(92.9)** | **13(92.9)** | **12(85.7)** | **7(50.0)** | **4(28.6)** | **13(92.9)** | **13(92.9)** |
| $bla_{CTX-M}+bla_{TEM}+bla_{SHV}$ | 1(3.33) | 1(100.0) | 1(100.0) | 1(100.0) | 1(100.0) | 0(0.0) | 0(0.0) | 1(100.0) | 1(100) |

Key: CXM-Cefuroxime, CAZ-Ceftazidime, AMC- Amoxicillin-clavulanic acid, CIP-Ciprofloxacin, CN- Gentamycin, SXT-Trimethoprim-sulfamethoxazole, N-Total sample screened, n-Number obtained.

isolates displaying an ESBL-phenotype among those recovered from diarrhoeic children in Kano found here was 12.8%, which is in the range of values found in diarrhoeic children in other African countries like Kenya 12% [22] and Libya 13.4% [23]. Interestingly, very different values were found in isolates also retrieved from diarrhoeic children in Iran [24] (25.9%), China [5] (5.6%) or USA [25] (7%). In developing countries, patients often receive antibiotics treatment without antibiotic susceptibility testing or prescription which will exert a selective pressure on the existing *E. coli*, whereas in developed countries strategies for reducing antimicrobial use have been put in place [24]. *E. coli* recovered from female subjects had a significantly higher probability of ESBLs production, although the reason for this difference could not be established in this study. However, this result is in agreement with previous reports by Vatopoulos *et al.* [26] that found that *E. coli* bearing transferable Resistance plasmids was more often associated with antibiotic resistance among female than male. ESBLs positive strains from our study were highly resistant to trimethoprim/sulphamethoxazole, which is similar to the reports of Miao *et al.* [27] and Valenza *et al.* [28]. All phenotypic ESBLs producing strains were susceptible to imipenem in agreement with reports from Tanzania [29], Libya [23], China [5, 27], and in contrast with the study of Hamprecht *et al.* [30] in Germany. The full susceptibility of isolates to imipenem may be attributable to the relatively low or non-usage of carbapenem drugs among the population. All the ESBLs positive isolates (genotypic) were resistant to tetracycline, similar to the report from Egypt [31] and Libya [23] (88.9% resistance), what could be due to an extensive use of this drug among human and animals.

$bla_{CTX-M}$ was very prevalent among the ESBL-positive isolates, in agreement with previous studies suggesting this gene is widespread worldwide [5, 7, 23, 27, 32–35], although $bla_{TEM-1}$ was also found in approximately three quarter of all ESBL isolates tested. No data on the presence of ESBL associated genes was available for clinical isolates in Nigeria; however $bla_{CTX-M}$ and $bla_{TEM}$ were also the predominant ESBLs associated genes in *E. coli* recovered from 200 cattle and 150 pigs in Nigeria [36]. Our detection rate is in the range of values reported from Egypt [31] (73.7% $bla_{CTX-M}$), Turkey [37] (73.43% $bla_{TEM}$) and USA [25] (74% $bla_{CTX-M}$). This study also shows that the carriage of multiple *bla* genes complicates the phenotypic interpretation of the resistance phenotypes, which is related to complex antimicrobial resistance [38]. Isolates with multiple combinations of *bla* genes and especially those carrying $bla_{CTX-M}+bla_{TEM}$ and $bla_{CTX-M}+bla_{TEM}+bla_{SHV}$ were resistant to a larger number of β-lactams (>92% resistance). $bla_{CTX-M}$ in combination with $bla_{TEM}$ gene was found in 46.7% of the isolates, similar to the findings of Harada *et al.* [39], which reported 48.8% of *E. coli* carried $bla_{CTX-M}+bla_{TEM}$ in clinical isolates from Japanese tertiary hospital, but higher than the result of Tawfick *et al.* [31] who found only 21.7% of *E. coli* isolates from diarrhoeic stool in Egypt carrying both $bla_{CTX-M}$ and $bla_{TEM}$.

In summary, our study demonstrates the presence of both phenotypically and genotypically ESBLs *E. coli* positive isolates in diarrheic children in Kano. This is a matter of concern since their presence was associated with high levels of phenotypic resistance. Performing antibiotic susceptibility testing on clinical isolates before antibiotic prescription could help to mitigate the emergence of antimicrobial resistance, and therefore regulations helping to control the sale of antibiotics by patent medicine stores and prudent antibiotic usage by clinicians is urgently needed in Nigeria.

## Supporting information

**S1 Table.**
(XLSX)

## Acknowledgments

We would like to thank the parent/guardian of the children who participated in this study. We acknowledge the technical support of Maria Garcia and Nisrin Maasoumi of Visavet Health Surveillance Centre in Madrid for their assistance during molecular analysis. We would also like to appreciate Dr. Fani F.E. for reading the first draft of this paper.

## Author Contributions

**Conceptualization:** Habeeb Kayode Saka, Nasir Tukur Dabo, Bashir Muhammad.

**Formal analysis:** Habeeb Kayode Saka.

**Funding acquisition:** Julio Alvarez.

**Investigation:** Julio Alvarez.

**Methodology:** Habeeb Kayode Saka, Silvia García-Soto, Julio Alvarez.

**Project administration:** Habeeb Kayode Saka.

**Supervision:** Nasir Tukur Dabo, Bashir Muhammad, Maria Ugarte-Ruiz, Julio Alvarez.

**Writing – original draft:** Habeeb Kayode Saka.

**Writing – review & editing:** Habeeb Kayode Saka, Vicente Lopez-Chavarrias, Julio Alvarez.

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
