## [Decision Letter · Decision Letter 0]

14 Feb 2020

PONE-D-19-35058

Molecular Detection of Extended Spectrum β-lactamase Genes in Escherichia coli clinical isolates from Diarrhoeic Children in Kano, Nigeria

PLOS ONE

Dear Dr. SAKA,

Thank you for submitting your manuscript to PLOS ONE. After careful consideration, we feel that it has merit but does not fully meet PLOS ONE’s publication criteria as it currently stands. Therefore, we invite you to submit a revised version of the manuscript that addresses the points raised during the review process.

We would appreciate receiving your revised manuscript by Mar 30 2020 11:59PM. To enhance the reproducibility of your results, we recommend that if applicable you deposit your laboratory protocols in protocols.io, where a protocol can be assigned its own identifier (DOI) such that it can be cited independently in the future. For instructions see: http://journals.plos.org/plosone/s/submission-guidelines#loc-laboratory-protocols

We look forward to receiving your revised manuscript.

Kind regards,

Monica Cartelle Gestal, PhD

Academic Editor

PLOS ONE

Journal Requirements:

Reviewers' comments:

Reviewer's Responses to Questions

**Comments to the Author**

1. Is the manuscript technically sound, and do the data support the conclusions?

Reviewer #1: Yes

Reviewer #2: Yes

2. Has the statistical analysis been performed appropriately and rigorously? 

Reviewer #1: Yes

Reviewer #2: Yes

3. Have the authors made all data underlying the findings in their manuscript fully available?

Reviewer #1: Yes

Reviewer #2: Yes

4. Is the manuscript presented in an intelligible fashion and written in standard English?

Reviewer #1: Yes

Reviewer #2: Yes

5. Review Comments to the Author

Reviewer #1: The study presents important data for local epidemiology. However, the absence of sequencing allows interpretation bias and reduces the impact of the results.

1.- Not all TEM variants are ESBL. Therefore, sequencing the complete orf is essential to identify the variant before relating its presence to the ESBL phenotype, especially in strains where only this gene was identified by PCR.

2. From the epidemiological point of view it is important the establishment of the variant of each investigated gene and the ST of at least all ESBL isolates.

Additionally, I strongly recommend discussing the significant difference between the presence of E. coli ESBL in men and women.

Reviewer #2: Dear author,

Please review next little details

38 E. coli, diarrhoeagenic E. coli (DEC)  could you defined it? As far as I know DEC does not exist, maybe EAEC, EHEC or EPEC.

42 Newer β-lactamase-producing enterobacteriaceae have been

43 isolated from clinical settings in different parts of the world (Bai et al., 2017) carrying factors such

44 as plasmid-mediated cephamycinases, extended spectrum β-lactamases (ESBLs), and

45 carbapenemases (Jacoby & Munoz-Price, 2005).

What do you mean?

They are inhibited by classical β- lactamase inhibitors such as

48 clavulanic acid, sulbactam and tazobactam (Pitout, 2013; Reuland et al., 2013).

Please upgrade with new inhibitors and combinations

99 20% glycerol were cultured onto EMB

100 agar at 37°C for 18-24 hours

Could you define the medium? ingredients or composition?

Table 1: Antibiotic Susceptibility 143 Pattern of ESBLs Producing E. coli

curiosity.....

what do you think or consider that can be the cause of this dis-match between phe- and geno-type

did you have some false negative??

Please correct some errors among both tables (1 and 2), CTX, CIP, STX and MDR not match the n (%)

f.ex CTX = 22(73.3) and table 2 sum 23

6. PLOS authors have the option to publish the peer review history of their article (what does this mean?). If published, this will include your full peer review and any attached files.

Reviewer #1: Yes: David Ortega-Paredes

Reviewer #2: Yes: Torres-Sangiao, E

---

## [Author Response · Author response to Decision Letter 0]

12 Jun 2020

Dear Editor,

We would like to thank the reviewers for their careful consideration of our manuscript. We have modified the paper according to their suggestions, and detail responses follow:

Reviewer#1

1.- Not all TEM variants are ESBL. Therefore, sequencing the complete orf is essential to identify the variant before relating its presence to the ESBL phenotype, especially in strains where only this gene was identified by PCR.

We agree with the point made by the reviewer, and in order to follow the suggestion we have now sequenced the complete blaTEM gene in the isolates that were positive in the PCR. This is now indicated in material and methods (lines 119-127) and results (lines 136-139). 

2. From the epidemiological point of view it is important the establishment of the variant of each investigated gene and the ST of at least all ESBL isolates.

We have identified the subtypes of the blaTEM genes identified in this study, but unfortunately the determination of the sequence type of the studied isolates is beyond the scope of the study and we have not been able to do so due to lack of resources. 

Additionally, I strongly recommend discussing the significant difference between the presence of E. coli ESBL in men and women.

We thank the reviewer for the comment, we have added a comment on gender differences following the reviewer’s suggestions in lines 169-173

Reviewer#2

Line 38: E. coli, diarrhoeagenic E. coli (DEC)  could you defined it? As far as I know DEC does not exist, maybe EAEC, EHEC or EPEC.

We appreciate the reviewers comment on this, DEC was designated as a group by Kaper and Nataro, 1998 (Nataro JP, Kaper JB. Diarrheagenic Escherichia coli. Clin Microbiol Rev. (1998)

11:142–201. doi: 10.1128/CMR.11.1.142) and since their pronouncement of DEC as a group, several articles and published books had recognized DEC as bacterial group.

Lines 42-45: Newer β-lactamase-producing enterobacteriaceae have been isolated from clinical settings in different parts of the world (Bai et al., 2017) carrying factors such as plasmid-mediated cephamycinases, extended spectrum β-lactamases (ESBLs), and carbapenemases (Jacoby & Munoz-Price, 2005). What do you mean?

We appreciate the comment of the reviewer. We are reviewing the antibiotic resistance trend in the β–lactam agents. We highlight how resistance move from penicillinase to β-lactamases before the newer β-lactamases. We have completed the sequence to clarify our point in lines 136-139.

Lines 47-48: They are inhibited by classical β- lactamase inhibitors such as clavulanic acid, sulbactam and tazobactam (Pitout, 2013; Reuland et al., 2013). Please upgrade with new inhibitors and combinations

We appreciate the comment of the reviewer, newer inhibitors now included along with a more updated reference in Line 48

Line 99: 20% glycerol were cultured onto EMB agar at 37°C for 18-24 hours. Could you define the medium? ingredients or composition?

Thank you for the comment. EMB medium is a selective/differential enteric media containing eosin and methylene-blue. The full name has been added to the manuscript line 94-95. 

Table 1: Antibiotic Susceptibility 143 Pattern of ESBLs Producing E. coli. curiosity..... what do you think or consider that can be the cause of this dis-match between phe- and geno-type. did you have some false negative??

The phenotypic identification of the ESBLs producing isolates was conducted in Nigeria, due to some challenges beyond our control, we could not recover 8 of the phenotypic ESBLs positive when the isolates were due to be transported to Madrid. Therefore, even though there were 38 ESBL-producing isolates (according to their phenotype) only 30 could be subjected to the PCRs for detection of the genes, and all of them were positive for at least one of them.

Please correct some errors among both tables (1 and 2), CTX, CIP, STX and MDR not match the n (%). f.ex CTX = 22(73.3) and table 2 sum 23

We appreciate the reviewer for the comment, we have double checked the tables all errors corrected

---

## [Decision Letter · Decision Letter 1]

9 Jul 2020

PONE-D-19-35058R1

Molecular Detection of Extended Spectrum β-lactamase Genes in Escherichia coli clinical isolates from Diarrhoeic Children in Kano, Nigeria

PLOS ONE

Dear Dr. Kayode Saka,

Thank you for submitting your manuscript to PLOS ONE. After careful consideration, we feel that it has merit but does not fully meet PLOS ONE’s publication criteria as it currently stands. Therefore, we invite you to submit a revised version of the manuscript that addresses the points raised during the review process.

We look forward to receiving your revised manuscript.

Kind regards,

Monica Cartelle Gestal, PhD

Academic Editor

PLOS ONE

Reviewers' comments:

Reviewer's Responses to Questions

**Comments to the Author**

1. If the authors have adequately addressed your comments raised in a previous round of review and you feel that this manuscript is now acceptable for publication, you may indicate that here to bypass the “Comments to the Author” section, enter your conflict of interest statement in the “Confidential to Editor” section, and submit your "Accept" recommendation.

Reviewer #2: (No Response)

2. Is the manuscript technically sound, and do the data support the conclusions?

Reviewer #2: Yes

3. Has the statistical analysis been performed appropriately and rigorously? 

Reviewer #2: N/A

4. Have the authors made all data underlying the findings in their manuscript fully available?

Reviewer #2: Yes

5. Is the manuscript presented in an intelligible fashion and written in standard English?

Reviewer #2: Yes

6. Review Comments to the Author

Reviewer #2: Dear authors,

I have no comments to add, though my point of view the I have recommend to publish this manuscript as letter, not as original report.

I hope you understand

My best

7. PLOS authors have the option to publish the peer review history of their article (what does this mean?). If published, this will include your full peer review and any attached files.

Reviewer #2: No

---

## [Author Response · Author response to Decision Letter 1]

6 Aug 2020

Dear Editor

Thank you for the kind consideration of our manuscript.

---

## [Decision Letter · Decision Letter 2]

13 Oct 2020

PONE-D-19-35058R2

Molecular Detection of Extended Spectrum β-lactamase Genes in Escherichia coli clinical isolates from Diarrhoeic Children in Kano, Nigeria

PLOS ONE

Dear Dr. SAKA,

Thank you for submitting your manuscript to PLOS ONE. After careful consideration, we feel that it has merit but does not fully meet PLOS ONE’s publication criteria as it currently stands. Therefore, we invite you to submit a revised version of the manuscript that addresses the points raised during the review process.

We look forward to receiving your revised manuscript.

Kind regards,

Monica Cartelle Gestal, PhD

Academic Editor

PLOS ONE

Journal Requirements:

Additional Editor Comments (if provided):

Reviewers' comments:

Reviewer's Responses to Questions

**Comments to the Author**

1. If the authors have adequately addressed your comments raised in a previous round of review and you feel that this manuscript is now acceptable for publication, you may indicate that here to bypass the “Comments to the Author” section, enter your conflict of interest statement in the “Confidential to Editor” section, and submit your "Accept" recommendation.

Reviewer #3: All comments have been addressed

2. Is the manuscript technically sound, and do the data support the conclusions?

Reviewer #3: Partly

3. Has the statistical analysis been performed appropriately and rigorously? 

Reviewer #3: Yes

4. Have the authors made all data underlying the findings in their manuscript fully available?

Reviewer #3: Yes

5. Is the manuscript presented in an intelligible fashion and written in standard English?

Reviewer #3: Yes

6. Review Comments to the Author

Reviewer #3: Interesting document, however, some recommendations are submitted

Comments:

1. Line- 87-88: The double disc synergy is not recommended by CLSI to confirm the presence o ESBL enzymes in Enterobacterales.

2.In table number 1. The Augmentin is a commercial name of amoxicillin-clavulanic acid. It should be changed.

3.- The characterization of CTX-M Beta-lactamases in humans samples must be complemented with DNA sequencing.

7. PLOS authors have the option to publish the peer review history of their article (what does this mean?). If published, this will include your full peer review and any attached files.

Reviewer #3: No

---

## [Author Response · Author response to Decision Letter 2]

15 Oct 2020

Dear Editor.

We would like to thank the reviewer for the careful consideration of our manuscript. We have modified the manuscript according to the suggestions and response follow:

Reviewer #3

1. Line- 87-88: The double disc synergy is not recommended by CLSI to confirm the presence o ESBL enzymes in Enterobacterales.

We appreciate the comment of the reviewer, the information now corrected to reflect our choice of using the double disc synergy test for more specificity in the confirmation of ESBLs phenotype in E. coli.

2. In table number 1. The Augmentin is a commercial name of amoxicillin-clavulanic acid. It

should be changed.

Thank you for the comment, we have made the correction.

3. The characterization of CTX-M Beta-lactamases in humans samples must be complemented

with DNA sequencing.

We agree with the submission of the reviewer, we could not go beyond typing blaTEM genes due to financial constraints.

---

## [Editor Report · Decision Letter 3]

17 Nov 2020

Molecular Detection of Extended Spectrum β-lactamase Genes in Escherichia coli clinical isolates from Diarrhoeic Children in Kano, Nigeria

PONE-D-19-35058R3

Dear Dr. SAKA,

We’re pleased to inform you that your manuscript has been judged scientifically suitable for publication and will be formally accepted for publication once it meets all outstanding technical requirements.

Kind regards,

Monica Cartelle Gestal, PhD

Academic Editor

PLOS ONE
---

## [Editor Report · Acceptance letter]

24 Nov 2020

PONE-D-19-35058R3 

Molecular Detection of Extended Spectrum β-lactamase Genes in *Escherichia coli* Clinical Isolates from Diarrhoeic Children in Kano, Nigeria. 

Dear Dr. Saka:

I'm pleased to inform you that your manuscript has been deemed suitable for publication in PLOS ONE. Congratulations! Your manuscript is now with our production department. 

Kind regards, 

on behalf of

Dr. Monica Cartelle Gestal 

Academic Editor

PLOS ONE